# Metabolomic Analysis Points to Bioactive Lipid Species and Acireductone Dioxygenase 1 (ADI1) as Potential Therapeutic Targets in Poor Prognosis Endometrial Cancer

**DOI:** 10.3390/cancers14122842

**Published:** 2022-06-08

**Authors:** Sònia Gatius, Mariona Jove, Cristina Megino-Luque, Manel Albertí-Valls, Andree Yeramian, Nuria Bonifaci, Miquel Piñol, Maria Santacana, Irene Pradas, David Llobet-Navas, Reinald Pamplona, Xavier Matías-Guiu, Núria Eritja

**Affiliations:** 1Oncologic Pathology Group, Department of Basic Medical Sciences, Biomedical Research Institute of Lleida (IRBLleida), University of Lleida, Av. Rovira Roure 80, 25198 Lleida, Spain; cmegino@irblleida.cat (C.M.-L.); malberti@irblleida.cat (M.A.-V.); ayeramian@irblleida.cat (A.Y.); nbonifaci@idibell.cat (N.B.); mpinolr.lleida.ics@gencat.cat (M.P.); fjmatiasguiu.lleida.ics@gencat.cat (X.M.-G.); 2Centro de Investigación Biomédica en Red de Cáncer (CIBERONC), Monforte de Lemos 3–5, 28029 Madrid, Spain; msantacana@irblleida.cat (M.S.); dllobet@idibell.cat (D.L.-N.); 3Department of Experimental Medicine, Biomedical Research Institute of Lleida (IRBLleida), University of Lleida, Av. Rovira Roure 80, 25198 Lleida, Spain; mariona.jove@udl.cat (M.J.); ipradas@irblleida.cat (I.P.); reinald.pamplona@mex.udl.cat (R.P.); 4Scientific and Technical Service of Immunohistochemistry, Biomedical Research Institute of Lleida (IRBLleida), Hospital Universitari Arnau de Vilanova, Av. Rovira Roure 80, 25198 Lleida, Spain; 5Molecular Mechanisms and Experimental Therapy in Oncology-Oncobell Program, Bellvitge Biomedical Research Institute (IDIBELL), Gran via De l’Hospitalet 199, 08908 L’Hospitalet de Llobregat, Spain; 6Department of Pathology, Hospital Universitari de Bellvitge, IDIBELL, University of Barcelona, Av. Gran via de l’Hospitalet 199, 08908 L’Hospitalet de Llobregat, Spain; 7Oncologic Pathology Group, Department of Medicine, Biomedical Research Institute of Lleida (IRBLleida), University of Lleida, Av. Rovira Roure 80, 25198 Lleida, Spain

**Keywords:** serous endometrial cancer, metabolomics profile, bioactive lipid species, ADI1, endometrial cancer

## Abstract

**Simple Summary:**

Uterine serous carcinoma is considered a rare and aggressive variant of endometrial cancer that accounts for 10% of all endometrial cancers diagnosed but is responsible for 40% of endometrial cancer-related deaths. Unfortunately, current treatments for serous endometrial carcinoma are ineffective. Therefore, there is a need to find new therapeutic targets. The aim of this study was to analyse the metabolic profile of serous cancer in order to identify new molecules and thereby define potential therapeutic targets. We observed that most of the differential metabolites are lipid species (suggesting the important role of the lipid metabolism). In addition, we found an increase in 2-Oxo-4-methylthiobutanoic acid (synthesised by the ADI1 enzyme) in serous carcinomas. Using public database analysis and immunohistochemistry, we established a correlation between elevated ADI1 levels and serous carcinoma. Furthermore, the ectopic modification of ADI1 expression in vitro revealed the ability of ADI1 to induce pathological cell migration and invasion capabilities.

**Abstract:**

Metabolomic profiling analysis has the potential to highlight new molecules and cellular pathways that may serve as potential therapeutic targets for disease treatment. In this study, we used an LC-MS/MS platform to define, for the first time, the specific metabolomic signature of uterine serous carcinoma (SC), a relatively rare and aggressive variant of endometrial cancer (EC) responsible for 40% of all endometrial cancer-related deaths. A metabolomic analysis of 31 ECs (20 endometrial endometrioid carcinomas (EECs) and 11 SCs) was performed. Following multivariate statistical analysis, we identified 232 statistically different metabolites among the SC and EEC patient samples. Notably, most of the metabolites identified (89.2%) were lipid species and showed lower levels in SCs when compared to EECs. In addition to lipids, we also documented metabolites belonging to amino acids and purine nucleotides (such as 2-Oxo-4-methylthiobutanoic acid, synthesised by acireductone dioxygenase 1 (ADI1) enzyme), which showed higher levels in SCs. To further investigate the role of ADI1 in SC, we analysed the expression protein levels of ADI1 in 96 ECs (67 EECs and 29 SCs), proving that the levels of ADI1 were higher in SCs compared to EECs. We also found that ADI1 mRNA levels were higher in p53 abnormal ECs compared to p53 wild type tumours. Furthermore, elevated ADI1 mRNA levels showed a statistically significant negative correlation with overall survival and progression-free survival among EEC patients. Finally, we tested the ability of ADI1 to induce migration and invasion capabilities in EC cell lines. Altogether, these results suggest that ADI1 could be a potential therapeutic target in poor-prognosis SCs and other Ecs with abnormal p53 expression.

## 1. Introduction

Endometrial cancer (EC) is the most common gynecologic malignancy worldwide, with approximately 417,367 newly diagnosed cases and 97,370 deaths in 2020 [1]. Additionally, the incidence of endometrial carcinoma is estimated to increase by 1–2% yearly [2]. EC is a disease that predominantly afflicts postmenopausal women (who represent 90% of patients diagnosed with EC cancer). However, 10% of cases are diagnosed in premenopausal women (5% of whom are under the age of 40) [3]. Although most ECs are diagnosed at early stages and cured with surgery alone, nearly 20% of EC cases recur after surgery to vaginal or pelvic regions, often metastasizing to distant sites [4]. The EC mortality rate relies on a subset of parameters and features present at diagnosis, including histological type, advanced stage and/or grade or lympho-vascular invasion. Moreover, those patients showing high-risk tumour features at diagnosis receive adjuvant radiation, chemotherapy or both. However, these treatments are estimated to benefit only 10–15% of all patients and, consequently, the median survival rates of patients with recurred or advanced EC are poor, with rates of less than one year [5].

Establishing the histological type of EC has consistently proven to be an important predictor of survival and the extent of the initial surgical procedure. Classically, histological and pathological evaluation classifies EC into endometrioid (EEC) and non-endometrioid (NEEC) tumours. EECs account for about 85% of EC cases and are usually associated with peri-menopausal and post-menopausal women, estrogen exposition, obesity and endometrial hyperplasia. Generally, EEC tumours have good prognoses [6]. In contrast, NEECs are very aggressive, oestrogen-independent tumours that usually arise in endometrial polyps in atrophic endometrium at older ages and present worse prognoses than EEC [7]. Serous carcinoma (SC) is the prototype of NEECs and accounts for less than 10% of all ECs; however, SC represents ≈40% of total EC deaths [8]. A new classification of EC from The Cancer Genome Atlas (TCGA) has provided additional (and potentially superior) prognosis information to traditional histologic typing and grading. Thus, through integrative genomic exome sequence analysis, the TCGA has revealed the existence of four different groups of EC tumours: Group 1, comprising EECs with mutations in the exonuclease domain of the DNA polymerase epsilon (POLE) and associated with good prognosis; group 2, including EECs with microsatellite instability; group 3, including EECs with low copy-number alterations (both groups 2 and 3 show similar progression-free survival rates, halfway between groups 1 and 4); and group 4 (copy-number high or serous-like), comprising most endometrial serous cancers and one quarter of the grade 3 endometrioid cases, with high numbers of gene copy-number alterations and p53 mutations and worse prognoses [9].

However, this molecular classifier does not replace clinicopathologic risk assessment and today it is thought that the integration of molecular classification into pathologic diagnosis may be important in the context of improving the prognosis and clinical management of EC patients [10,11].

Extensive research using high-throughput technologies including proteomics, transcriptomics and genomics has increased the characterisation of molecular changes at different levels of cellular expression specifically associated with human endometrium and the onset and progression of EC [12,13]. Metabolomics is believed to have the potential to reflect the phenotype of the cell more precisely and to provide more fundamental and global information than genomics, proteomics and transcriptomics. Metabolomics determines the end point of cellular processes and hence defines the readout of the current physiological status of the system. For instance, discoveries related to oncometabolites have highlighted the potential unsuspected cellular pathways, the components of which could serve as diagnosis and prognosis biomarkers or as possible therapeutic targets for disease treatment [14].

Still, very few studies have used metabolomics in the context of EC and most of them have been conducted comparing non-tumour tissue or plasma from control subjects with EEC patients [15,16,17]. In the present study, for the first time, metabolomic profiling was determined using an LC-MS/MS platform to systematically define specific phenotypic patterns associated with histological types of endometrial cancer. More specifically, the main objective of the present study was to evaluate metabolomic differences between SC and EEC tissues in order to identify potential biomarkers for differential diagnosis and prognosis as well as metabolic molecules and pathways involved in the development and progression of SC. Here, we identified a particular metabolic signature related to SC and delved into the mechanistic basis of ADI1 overexpression as a new potential therapeutic target in endometrial carcinogenesis.

## 2. Materials and Methods

### 2.1. Patient Sample Collection

All human samples were from patients diagnosed and treated at the Hospital Universitari Arnau de Vilanova de Lleida (HUAV). Tumour samples are available at IRBLLEIDA Biobank registered in the National Registry of Biobanks of Instituto de Salud Carlos III (B.00682). All samples were collected following the current regulations in the biomedical research law, the Royal Decree of Biobanks RD 1716/2011, the European Regulation 2016/679 and the organic law LOPD-GDD 3/2018 on the protection of personal data. Clinical follow up of all patients was obtained from HUAV Clinical Histories. Duration of follow up was defined as time from diagnosis to last date follow up found in clinical history of each patient. Four criteria were defined depending on the clinical status of each patient in last date follow up: (1) No evidence of disease (NED), (2) alive with disease (AWD), (3) dead of disease other than endometrial cancer (DAD) and (4) dead of endometrial disease (DOD). Presence of recurrence during follow up was also collected.

### 2.2. Metabolite Extraction

Tissue samples were homogenised in a buffer containing 180 mM KCl, 5 mM 3-[N-morpholino] propanesulfonic acid, 2 mM ethylenediaminetetraacetic acid (EDTA), 1 mM diethylenetriaminepentaacetic acid and 1 mM butylated hydroxyl toluene, 10 mg/mL aprotinin, 1 mM phenylmethylsulfonyl fluoride, pH 7.3 with a Potter–Eljeveim device (DWK Life Sciences GmbH, Hattenbergstr, Germany) at 4 °C. The protein concentration was measured using the Lowry assay (Bio-Rad Laboratories, München, Germany) with bovine serum albumin used as a standard. In the case of each sample, 100 μg were resuspended in a total of 30 μL of homogenising buffer and vortexed for one minute. In order to extract metabolites from tissue homogenates, 90 μL of ice-cold methanol were added to each sample, and the samples were incubated at −20 °C for 1 h and centrifuged at 12,000× *g* for 3 min at room temperature, as described elsewhere [15]. The supernatants were recovered, evaporated using a Speed Vac (Thermo Fisher Scientific, Barcelona, Spain) and resuspended in water containing 0.4% acetic acid.

### 2.3. Metabolome Analysis

For the metabolomic analysis, an Agilent 1290 LC system coupled to an ESI-Q-TOF MS/MS 6520 instrument (Agilent Technologies, Barcelona, Spain) was used. In all cases, 2 μL of extracted sample were applied onto a reversed-phase column (Zorbax SB-Aq 1.8 μm 2.1 × 50 mm; Agilent Technologies, Barcelona, Spain) equipped with a precolumn (Zorba-SB-C8 Rapid Resolution Cartridge 2.1 × 30 mm 3.5 μm; Agilent Technologies, Barcelona, Spain) with a column temperature of 60 °C. The flow rate was 0.6 mL/min. Solvent A was composed of water containing 0.2% acetic acid, and solvent B was composed of methanol 0.2% acetic acid. The gradient started at 2% B and increased to 98% B in 13 min and held at 98% B for 6 min. Pot-time was set at 5 min. Data were collected in positive electrospray mode, with the TOF operated in full-scan mode at 50–1600 m/z in an extended dynamic range (2 GHz) using N2 as the nebuliser gas (10 L/min, 350 °C). The capillary voltage was 4000 V, with a scan rate of 1.5 scan/s. The ESI source used a separate nebuliser for the continuous, low-level (10 L/min) introduction of reference mass compounds: 121.050873 and 922.009798 (positive ion mode) and 119.036320 and 966.000725 (negative ion mode), which were used for continuous, online mass calibration.

### 2.4. Metabolomic Data Analysis and Statistics

Data collection was performed using MassHunter data analysis software (Agilent Technologies, Barcelona, Spain), and the molecular characteristics of the compounds were obtained using MassHunter qualitative analysis software (Agilent Technologies, Barcelona, Spain). The molecular feature extractor (MFE) algorithm (Agilent Technologies, Barcelona, Spain) was used to represent the different ions with their own characteristics. Finally, MassHunter Mass Profiler Professional and Metaboanalyst software [18] were used for unsupervised metabolomics analysis. Only samples with at least two ions were selected. Multiple load states were excluded. Only compounds with common characteristics present in more than 50% of samples of the same condition were considered. Those masses that demonstrated significant differences (*t*-test, *p* ≤ 0.05) were searched for in METLIN Metabolite PCD/PCDL (Agilent Technologies, Barcelona, Spain). The identification of each metabolite was based on an orthogonal approach: identical chromatographic behaviour (RT + −0.5 min) and identical mass/charge (≤30 ppm) in those molecules present in the PCDL database. Furthermore, potential identities were also obtained by searching in the HBMD database [19], considering M + H, M + NH4, M + H-H_2_O and M + Na adducts. The identity of these metabolites was then confirmed, when possible, by the MS/MS spectrum obtained in different runs.

### 2.5. Bioinformatic Analysis of Clinical Data

Gene expression data (RSEM), gene mutational data and the related clinical information of TCGA-UCEC [20] patients were downloaded from cbioportal [21,22]. The Wilcoxon rank sum test (wilcox.test R function) was performed to test the difference in terms of expression levels of ADI1 between tumour subtypes and the TP53 mutation status. The Maxstat package [23] was used to determine a cutoff point to stratify patients into ADI1 high-expression and low-expression groups. Kaplan–Meier plots for progression-free survival and overall survival, adjusted to the Cox model of proportional hazards, were used to illustrate the survival curves of ADI1 for the two groups.

### 2.6. Inmunohistochemical Analysis

For the validation of the results of the metabolomic analysis with immunohistochemical techniques, two TMAs were constructed: one with endometrioid carcinomas and one with serous carcinomas. Two areas of each case were selected for TMAs from tumour samples.

TMAs were built with the 3DHistech TMA Grand Master automatic array and its corresponding software 3DHistech TMA Control software (3DHistech Limited, Budapest, Hungary). Each TMA was cut into 3 micron sections and dried at 65 °C. The deparaffinisation, rehydration and antigen recovery pretreatments were then performed in the PT-LINK pretreatment module (DAKO) at 95 °C for 20 min. with 50 × Tris/EDTA buffer at pH9. Endogenous peroxidase was inhibited before staining the sections.

Acireductone Dioxygenase 1 (ADI1) antibody was used. After incubation, the reaction was visualised with the EnVision Detection Kit (DAKO) using diaminobenzidine as substrate. Sections were counterstained with hematoxylin. Positive and negative controls were tested for each technique. Immunohistochemical assessment was performed by two pathologists using pre-established assessment criteria. Immunoexpression was graduated semiquantitatively considering the percentage of stained cells and the intensity of the staining. A score or histoscore from 0 (no immunostaining observed) to 300 (maximum immunostaining) was obtained from each of the TMA samples. The histocore of each case was calculated using the following formula: HISTOSCORE = 1 × (% of cells with mild intensity) + 2 × (% of cells with moderate intensity) + 3 × (% of cells with strong intensity).

### 2.7. Cell Culture

Human endometrioid EC cell HEC-1a and AN3CA, were purchased from the American Type Culture Collection (2008; ATCC-authentication by isoenzymes analysis). Ishikawa 3-H-12 (IK) and MFE-296 cell lines were purchased from Sigma-Aldrich (Saint Louis, MO, USA). Cells were expanded in Dulbecco’s modified Eagle’s medium (DMEM; Sigma-Aldrich) supplemented with 10% foetal bovine serum (FBS; Invitrogen), 1 mmol/L HEPES (Sigma-Aldrich), 1 mmol/L sodium pyruvate (Sigma-Aldrich), 2 mmol/L L-glutamine (Sigma-Aldrich), 1% penicillin/streptomycin (Sigma-Aldrich) at 37 °C with saturating humidity and 5% CO_2_.

### 2.8. Plasmid Transfection

The lentivirus carrying ADI1 shRNA clones were purchased from Applied Biological Materials Inc. The lentivirus transduction was conducted as previously described [24]. The clone IDs of ADI1 shRNAs used in this study were #i00613a and #i000613b. The target sequences were 5′-GAACTACTCCTGGATGGACAT-3′ and 5′-GAGCATTTGCACTTGGACGAT-3′, respectively. Plasmid for ADI1 overexpression (forward sequence 5′-AGTTAAGCTTATGGTGCAGGCCTGGTATAT-3′ inserted in pcDNA™3.1/V5-HisB-ADI) were kindly provided by Dr. Chu [25]. Plasmid transfection was performed with Lipofectamine 2000 reagent (Invitrogen Incorporated, Waltman, MA, USA) following the manufacturer’s instructions.

### 2.9. Wound Healing Assay

Cells were plated in 24-well plates and incubated to almost complete confluence, and a 10  μL pipette tip was then used to scratch the cell layer. The gap widths were imaged in an inverted microscope after scratching (0 h and 24 h) and these were measured using the Image J (NIH LOCI, University of Wisconsin, Madison, WI, USA) program.

### 2.10. Transwell Invasion Assay

Cells were plated in the upper chamber of the Transwell (8 µm pore, Corning Incorporated, Corning, NY, USA) coated with Matrigel in serum-free medium at a density of 1 × 10^4^ per well. FBS 10% was used as a chemoattractant. After 48 h, cells were fixed with paraformaldehyde 4% and stained with Hoechst (5 µg/mL). Finally, cells were pictured with an epifluorescence microscope (Leica, Wetzlar, Germany), before and after a cotton swab. Results were analysed to obtain the percentage of invasive cells using the software Image J.3 (NIH LOCI, University of Wisconsin, Madison, WI, USA). 

## 3. Results

### 3.1. Clinical Parameters of Human Samples

To conduct the metabolomic analysis, a total of 31 cases were selected, consisting of 20 samples of endometrial endometrioid carcinoma (EEC) of different histological grades and stages (Table 1) and 11 samples of serous carcinoma (SC) (high histological grade), previously frozen in liquid nitrogen. For the immunohistochemical study, 96 cases (with 67 being EEC and 29 SC) were selected. All samples were embedded in paraffin blocks. The series of endometrial endometrioid carcinomas consisted of 33 EEC grade I, 23 EEC grade II and 11 EEC grade III carcinomas. The pathological stage of all endometrial endometrioid carcinomas was determined. We identified 55 carcinomas in stage 1, 7 carcinomas in stage 2, 4 carcinomas in stage 3, and 1 in stage 4. The pathological stage of all serous carcinomas was determined and 10 carcinomas in stage 1, 3 carcinomas in stage 2, 9 carcinomas in stage 3 and 7 in stage 4 were identified (Table 2). To obtain good quality with respect to the inmunohistochemical analysis, the use of old FFPE tissue samples is not recommended [26,27,28]. As metabolomics analysis was developed prior to immunohistochemical validation, paired samples were not used for this study.

### 3.2. Characterisation of Metabolomic Differences between SC and EEC

The first aim of this work was to recognise metabolomic differences between SC and EEC. Low molecular weight ionisable molecules (m/z ≤ 3000) were selected for the analysis (5649). In order to define the most reliable and robust metabolites, we selected only those molecules which were present at least in 50% of the samples in any group (1037) and applied multivariate statistics. Principal component analysis (PCA) (Figure 1A) revealed a differential metabolomics profile in SC and EEC carcinomas, showing that SC samples are more homogenous than EEC samples. These differences were confirmed when hierarchical clustering analysis was applied to the same features (Appendix A). Additionally, when a supervised approach was performed, partial least discriminant analysis (PLS-DA) reinforced the idea of a specific metabolomics signature that can discern between SC and EEC (accuracy: 1, R2: 0.99, Q2: 0.59) (Figure 1B). Figure 1C shows the 15 metabolites that contribute most to define the first component of the PLS-DA, which better discriminates between groups (Figure 1C). All these metabolites, with the exception of the unknown compound 377.7853_11.7, were decreased in SC individuals. Then, with the objective to further describe the metabolomics signature, we selected the 25 lipid species with the lowest *p*-values, thus obtaining a perfect clusterisation of the samples according to their group (Figure 1D,E). Notably, 23 of the 25 metabolites were down-regulated in SC samples, whereas two of them (adenosine monophosphate and 2′-Deoxyguanosine 5′ monophosphate.) were increased in SC samples (Figure 1F).

Following multivariate statistics, we focused on defining the specific metabolites that are statistically different from one group to the other (Student *t*-test, *p* < 0.05). We found 232 metabolites, 54 of them with a corrected *p*-value (FDR) < 0.05 (Appendix A). Among the 232, 93 (40%) were identified as belonging to different families based on exact mass, retention time, isotopic distribution and/or MS/MS spectrum (Table 3). In particular, we found 3 (3.2%) benzene and substituted derivative metabolites, 5 (5.43%) carboxylic acids and derivatives, 7 (7.5%) fatty acyls (4 fatty acids and conjugates, 1 eicosanoid, 1 lineolic acid and derivative and 1 fatty acid ester), 15 (16.1%) glycerolipids (6 monoradylglycerols and 9 diradylglycerols), 46 (49.4%) glycerophospholipids (19 glycerophosphates, 10 glycerophosphoglycerols, 1 glycerophosphoinositol, 6 glycerophosphocholines, 6 glycerophosphoethanolamines, 2 glycerophosphoserines, 1 glycerophosphoinositolphosphate and 1 CDP-glycerol), 1 (1%) organoxigen compound, 2 (2%) prenol lipids, 1 (1%) purine nucleotide, 10 (10.7%) sphingolipids (all of them sphingomyelins) and 3 (3.2%) steroid and steroid derivatives. Notably, mostly lipid species (96.4%) showed lower levels in SC compared to EEC, while those metabolites that belonged to amino acids and purine nucleotides showed higher levels in SC. The regulation (low or high levels) for all the differential metabolites is shown in the Appendix A.

Globally, the characterisation of the metabolomic differences between EEC and SC showed a differential profile between these two histological subtypes defining a particular signature associated with specific metabolites. Among the identified metabolites, we found several lipids but also different non-lipid species with a potential role in the molecular characterisation of endometrial endometrioid carcinomas and serous carcinomas. We focused our attention on those metabolites up-regulated in SC because of its major aggressiveness and searched for specific antibodies to analyse the affectation of related enzymes.

### 3.3. ADI1 Is Overexpressed in SC Patients

In an attempt to further characterise and validate the metabolomic results obtained, we searched for available antibodies targeting specific enzymes. First, we were struck by the increased levels of two branched-chain amino acids (BCAAs) (valine and isoleucine) of the valine-leucine-isoleucine metabolism pathway in the SC tumours. Therefore, we decided to evaluate the involvement of this pathway in EC tumours. 

BCAA aminotransferase 1 (BCAT1) is the enzyme responsible for the synthesis and degradation of valine and leucine. In addition, BCAT1 is also involved in other pathways such as “cysteine and methionine metabolism”. To explore the role of this enzyme, an immunohistochemical evaluation of BCAT1 protein was performed. Unfortunately, the results obtained did not show significant differences when comparing protein expression levels in EEC and SC (data not shown).

Another metabolite identified as up-regulated in SC tumours was 2-Oxo-4-methylthiobutanoic acid. 2-Oxo-4-methylthiobutanoic acid is also related to “cysteine and methionine metabolism”, as it is a precursor of L-methionine, synthesised by acyreductadioxygenase 1 (ADI1 or also called Sip-L and MTCBP1). Therefore, we assessed ADI1 protein expression by immunohistochemistry in three independent TMAs. Consistent with previous metabolomics results, we observed a significant overexpression of ADI1 in SC samples compared to EEC samples (Figure 2A) (no differences were found in terms of grade and stage of patients at diagnosis). Next, we analysed the ADI1 mRNA levels from the publicly available TCGA_UCEC database which contains a large cohort of data from endometrial cancer patients [22]. Remarkably, we found that ADI1 mRNA expression levels were enriched in SC patients compared to EEC patients (Figure 2B). All these results reinforced the metabolomics data obtained before.

Given the observed increased ADI1 levels in SC patients compared to EEC patients, we decided to determine progression-free survival (PFS) rates in SC and EEC patients expressing high levels or low levels of ADI1. Log-rank analysis for PFS from a cohort including 130 patients diagnosed with SC showed a remarkable (although not statistically significant) tendency toward a negative correlation between high ADI expression levels and PFS (*p*-value = 0.06) rates (Figure 2C). Surprisingly, when analysing this correlation in a cohort of 396 EEC patients, we noticed that the PFS parameter presented a statistically significant correlation with high ADI expression levels (*p*-value = 0.01) (Figure 2D).

### 3.4. High ADI1 mRNA Levels Are Associated with Altered P53 Expression 

P53 is a tumour suppressor gene altered in ECs, which has been shown to be an independent indicator of poor prognosis [29]. Serous carcinomas are high grade tumours with bad prognosis and usually show altered p53 expression. However, it is also well known that some EECs can also present an altered expression of p53 [30]. This group of tumours are usually high-grade carcinomas and have bad behaviour. Thus, p53 expression has been frequently used as a surrogate marker in SC and EEC [8]. Therefore, we used the TCGA_UCEC database to analyse the association between ADI1 and P53 mutant expression in endometrial carcinomas. Interestingly, as shown in Figure 2E, we found a significant increased expression of ADI1 among those ECs with mutant P53 expression. Next, we further explored the association of ADI1 overexpression and P53 mutant status in EEC. In concordance with previous analyses, mutant p53 EEC also displayed significantly elevated mRNA levels of ADI compared to wildtype P53 expression in EEC (Figure 2F). 

### 3.5. Alteration of ADI1 Expression Modulates Migration and Invasion Capabilities of EEC Cell Lines through Regulation of Epithelial–Mesenchymal Transition Process

In an attempt to investigate the role of ADI1 in the acquisition of an aggressive phenotype in EEC, we performed a western blot assay to assess the levels of ADI1 in four EEC cell lines (Figure 3A). The results obtained showed that differentiated EC cell lines express moderately low levels of ADI1, whereas the dedifferentiated MFE-296 cell lines express high ADI1 protein levels. 

Next, we transfected ADI1 over-expression vectors into a HEC-1A cell line and it infected with lentiviral shRNA-ADI1 plasmids MFE-296 cells in order to analyse the functions of ADI1 (Figure 3B). Notably, by performing wound healing and transwell invasion assays, we found that the overexpression of ADI1 in HEC-1A cells significantly enhances the migratory and invasive capacities of these cells (Figure 3C,D). Conversely, lentivirus-mediated ADI1 downregulation led to the opposite effect in MFE-296 cells in terms of migratory and invasive competencies (Figure 3E,F). 

It is well known that epithelial–mesenchymal transition (EMT) changes are central processes in the migration and invasion phenomena of endometrial neoplastic cells [31]. Therefore, on the basis of the evidence observed, we next assessed the expression of several epithelial and mesenchymal markers on the ADI1 modulated-expression cell lines by immunofluorescence and western blot assays. Thus, the results obtained show that in the HEC-1A cell line over-expressing ADI1 presented a down-expression of epithelial protein markers such as E-cadherin and β-catenin and an up-regulation of the mesenchymal protein marker vimentin and the EMT transcriptional factor Snail (Figure 4A,B). On the other hand, ADI1 knockdown MFE-296 cells presented a raised expression of E-cadherin and β-catenin epithelial markers and down-regulated protein levels of vimetin and Snail (Figure 4C,D). 

## 4. Discussion

Uterine serous carcinomas are still considered a relatively rare tumour that accounts for about 10% of all ECs, although 40% deaths in EC cases are attributed to this EC subtype [8]. SC, by definition, is regarded as a high-grade tumour and has a high tendency to develop lymph node metastasis as well as adnexal and peritoneal spread, even in cases where the primary tumour is small [32]. The terminology “high-risk EC” has been adopted to include SC, clear cell carcinoma and grade 3 EEC due to the poor prognosis associated with these types of tumours. These tumours present low-rate mutations, but the p53 mutation could explain the poor prognosis observed, since the altered expression of p53 has been clearly documented to result in a high degree of genomic instability and fast tumour progression and invasion [33]. A number of previous metabolomics studies have been aimed at defining diagnosis, prognosis and predictive markers of EC. Two recent reviews, references [34,35], systematically summarise these findings. Until now, however, nothing was known about the metabolic differences between EEC and SC EC subtypes. The results presented in the present study suggest that some metabolic pathways are regulated differently depending on the type of EC tumour.

A relevant observation of the present study is that among the differential and identified metabolites that we have reported, 89.2% (83) are lipid species (most of them previously defined as bioactive lipid molecules), suggesting the important role of lipid metabolism in the phenotypic differences between SC and ECC. Remarkably, 96.4% of these lipid species showed lower levels in SC compared to EEC. Among the lipid species with increased levels in SC, the presence of two bile acids is noteworthy. It is well known that bile acids, which are also bioactive lipids, regulate different nuclear receptors including HIF-1alpha [36]. Within the lipid species with decreased levels in SC, we detected signalling molecules such as fatty acids and conjugates, eicosanoids, monoacylglycerols, diacylglycerols (DAG), lysophosphatidic acids, phosphatidic acids (PA), phosphatidylinositols (PI) and derivatives as well as one bile acid. The other identified lipid species are essentially structural components of the lipid bilayer and substrates for bioactive lipid species (phosphatidycholines, phosphatidylethanolamines and sphingomyelins). These bioactive lipids participate in the regulation of a broad spectrum of cellular processes such as inflammation and cell differentiation, growth, proliferation and survival, which are key in the biology of cancer. Particularly striking is the number of lipid species (31) belonging to the PI-DAG-PA signalling axis. DAG has unique functions as a basic component of membranes, an intermediate in lipid metabolism and a key element in lipid-mediated signalling, essential processes for cell homeostasis and survival, and its levels are strictly regulated [37]. Basically, DAGs are generated by the phospholipase C (PLC)-mediated hydrolysis of PI(4,5)P2 (phosphatidylinositol 4,5-bisphosphate), which is also the substrate for PI3K (phosphatidylinositol 3 kinase) activity. DAGs are phosphorylated in a reaction catalysed by diacylglycerol kinases (DGKs) to generate PA, which is the first step in a series of reactions that replenish PI(4,5)P2 levels in the PI cycle. Other DAG sources include the pool generated by phosphatidic acid phosphatases (PAPs) acting on the PA generated by the phospholipase D (PLD)-catalysed hydrolysis of PC (phosphatidylcholine) or that originate from ceramide and PC by the action of sphingomyelinases (SMS) [38]. In cells, DAG, PA and PI(4,5)P2, as lipid second messengers, play an important role in regulating several signal transduction proteins [39] that, in turn, are linked to the regulation of a broad diversity of functions, including the activation of protein kinases, the activation of ion channels and cytoskeletal reorganisation, among several others [38,39,40], which can promote endometrial cancer progression. In this context, the lower levels of these lipid signalling species could be counterintuitive with respect to their role in cell physiology and the cellular traits expressed by SC tissue. So, it is hypothesised that the lower content of these lipid species expresses a high consumption by the enzymes involved in their signalling pathways in order to sustain cell processes in SC cancer cells. Consistent with this, the high levels observed for specific amino acids as well as nucleotides in SC would indicate the high demands of protein biosynthesis and genetic material necessary to ensure the survival, growth, proliferation and invasion of cancer cells into SC tissue and would reinforce these insights when it comes to lipids. Further studies are needed, however, in order to consolidate these new insights.

In addition to lipids, we also documented a higher level of two BCAAs, valine and isoleucine, comparing SC to EEC. BCAAs are essential amino acids that are involved in protein biosynthesis, but they also have other functions; BCAA metabolism has recently been proposed as an oncogenic metabolic pathway involved in metabolic reprogramming and cancer progression in several human cancers such as breast cancer and ovarian cancer via various mechanisms including mTOR signalling, ROS homeostasis and the response to hypoxia [41]. In endometrial tumorigenesis, a previous study has shown that high levels of BCAAs and intermediates from BCAA pathways are associated with EC tumour aggressiveness and recurrence [42,43]. Furthermore, prior studies described different metabolites belonging to cysteine and the methionine metabolism as potential biomarkers for the diagnosis and prognosis of EC [44]. In this regard, we reported that 2-oxo-4-methylthiobutanoic, a metabolite of this pathway, was increased in SC. 

After analysing the results obtained through the application of high throughput metabolomics techniques, we also analysed the protein expression levels of the ADI1 enzyme. Our results showed that both SC and EEC with the mutant expression of p53 presented an increased expression of ADI1. The conserved role of ADI1 has been described and classified as an ARD/ARD′ (aci-reductone dioxygenase) member protein [45]. As an acireductone dioxygenase, ADI1 participates in the methionine salvage pathway or the 5′-methylthioadenosine (MTA) cycle and needs Fe^2+^ as a cofactor to achieve its function in the production of 2-oxo-4-methylthiobutanoic acid, a key step in this pathway [25]. In addition to serving as a key enzyme in the MTA cycle, the role of ADI1 in carcinogenesis is very controversial. On the one hand, ADI1 has been described as a potential tumour suppressor in hepatocellular carcinoma and prostate tumours [25,46,47]. On the other hand, increased ADI1 expression has been described in glioblastoma tumours and implicated in the acquisition of cellular migration capacity [48]. Although expressed in the uterus [49], the function of ADI1 and its involvement in endometrial carcinogenesis remain elusive. Here, we demonstrated how ADI1 is overexpressed in SC and mutant p53 EEC, and its increased expression results in a poor outcome for EEC patients. This could be explained by the alterations in migratory and invasive capabilities that determine the overexpression of ADI1 in endometrial carcinoma cells. While it is appealing to assume that the upregulation of ADI1 and its ability to induce EMT and malignant transformation in EC cells could be a consequence of p53 modified expression, as some studies have suggested [50,51], the molecular mechanism by which ADI1 induces the observed malignant transformation is unknown. One proposed model suggests that ADI1 may also play a role in the phosphorylation of casein kinase II protein (CK2), since three potential CK2 phosphorylation sites (ser10, Ser6 and Thr136) have been described [52]. This hypothesis could explain the observed results, as increased CK2 activity has been associated with cell proliferation and anchorage-independent growth in endometrial cancer cells [53]. Another explanation is that ADI1 abrogates MT1-MMP (membrane type-1 matrix metalloproteinase) signalling that triggers autophagy [48,54]. Thus, an unregulated autophagy process may enable EC cells to escape from cell death and may help them develop adaptive mechanisms to survive. Partial or complete EMT has been associated with the invasive phenotype in ECs [31], as the EMT process allows cells to move away from their epithelial cell community and to integrate into the surrounding tissue, even at remote locations. The results presented here show that cells over-expressing ADI1 show a decrease in epithelial protein markers and an increase in mesenchymal proteins. Moreover, these changes endow ADI1-upregulated cells with migratory and invasion capabilities, encompassing the invasive growth mode of the primary tumour and shifting it into a biologically aggressive state that leads to poor outcomes for EC patients.

## 5. Conclusions

The changes in metabolism observed in tumour cells suggest that the study of metabolomes in cancer tissue provides useful information to characterise both EEC and SC. Our metabolomic approach found that SC cancer cells showed lower levels of lipid signalling species compared to EEC cells, along with an increased content of specific amino acids and nucleotides, suggesting intensive protein and nucleotide biosynthesis as well as the activation of lipid-mediated signalling pathways likely to ensure the survival, growth, proliferation and invasion of cancer cells that characterise endometrial serous carcinomas. Furthermore, the present study provides evidence that ADI1 overexpression is associated with SC and mutant p53 EEC tumours and its increased expression confers poor outcomes in EEC patients. In this respect, further prospective studies or trials focusing on ADI1 expression in SC and p53 mutant EEC are needed to determine its potential value as a new therapeutic target in endometrial carcinogenesis. 

## Figures and Tables

**Figure 1 cancers-14-02842-f001:**
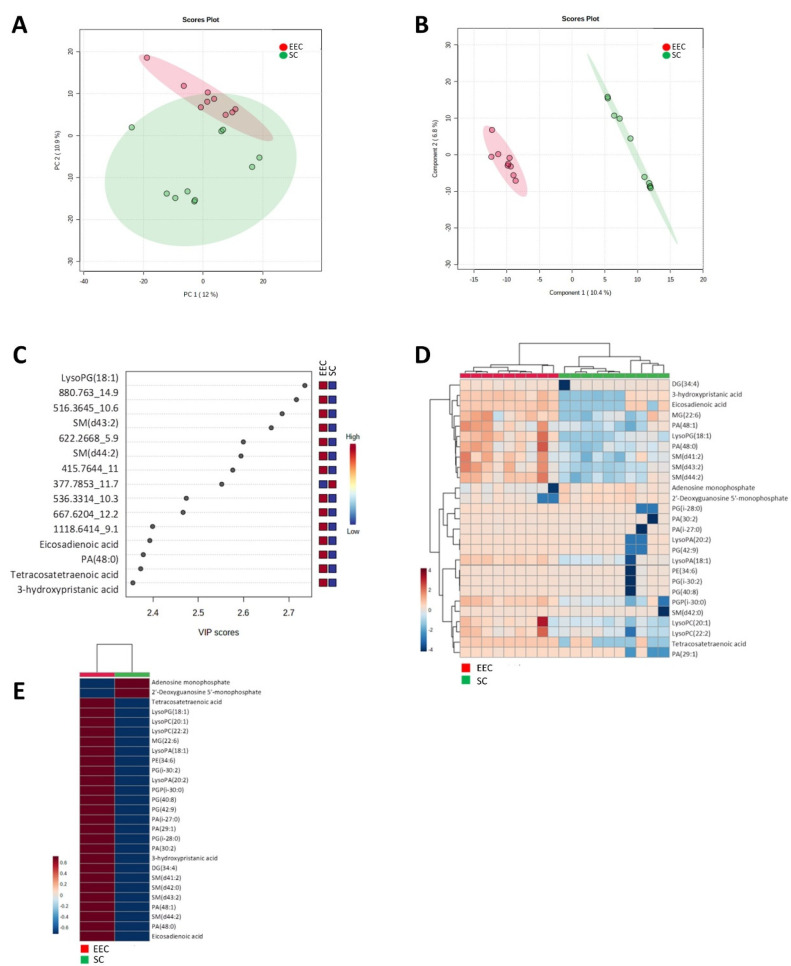
Multivariate statistics using a whole metabolome show a specific signature defining endometrioid and serous carcinomas. (**A**) Two-dimensional principal component analysis (PCA) representation of features using two principal components. (**B**) Hierarchical clustering of individual samples according to metabolite levels. (**C**) Two-dimensional partial least square discriminant analysis (PLS-DA) representation of features using two principal components (accuracy: 1, R2: 0.99, Q2: 0.59). (**D**) Variable important projection (VIP) scores indicating the elements which contribute most to define the first component of a PLS-DA. (**E**) Hierarchical clustering of individual samples according to metabolite level using those metabolites with a corrected *p*-value (FDR) < 0.05. F) Hierarchical clustering of average sample values according to metabolite level using those metabolites with a corrected *p*-value (FDR) < 0.05.

**Figure 2 cancers-14-02842-f002:**
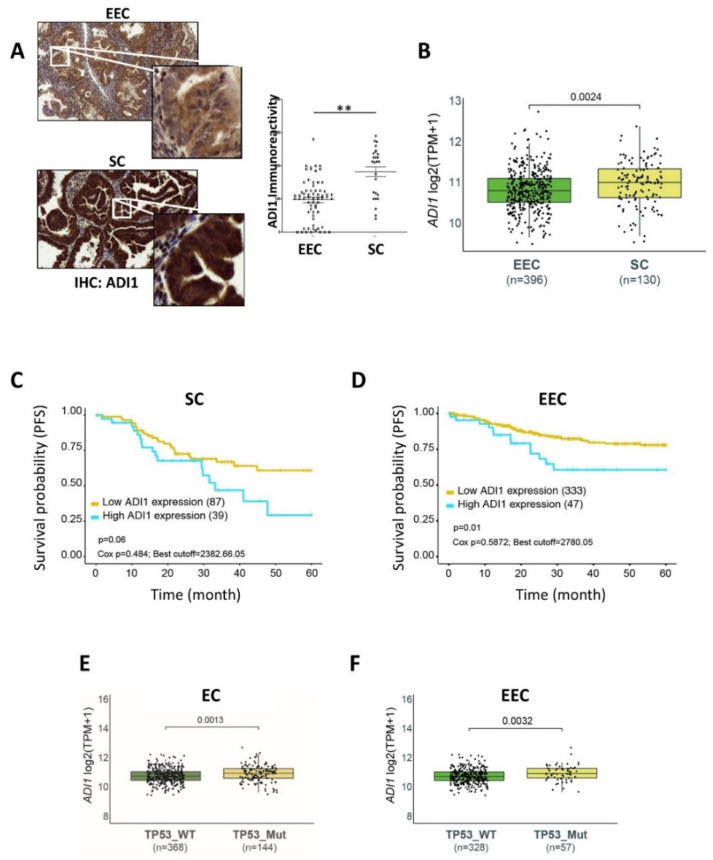
ADI1 expression and progression-free survival (PFS) of endometrial cancer patients. (**A**) Immunohistochemistry (IHC) analysis of ADI1 expression levels in EEC and SC samples. (Left) Representative images. Magnification 20× and 100×. (Right) Quantification graph. Graph value is the mean, and error bars are represented as mean ± S.E.M. Statistical analysis was performed using two-tailed paired Student *t*-test analysis. ** *p* < 0.01. (**B**) ADI1 mRNA expression levels in EEC and SC patients. (**C**) Kaplan–Meier plots of PFS of SC patients, stratified into ADI1 high-expression and low-expression groups. (**D**) Kaplan–Meier plots of PFS of EEC patients, stratified into ADI1 high-expression and low-expression groups. (**E**) ADI1 mRNA levels of EC patients stratified according to TP53 mutation status. (**F**) ADI1 mRNA levels of EEC samples stratified according to TP53 mutation status. Gene expression data, gene mutational data and their related clinical information were obtained from TCGA-UCEC public database [20].

**Figure 3 cancers-14-02842-f003:**
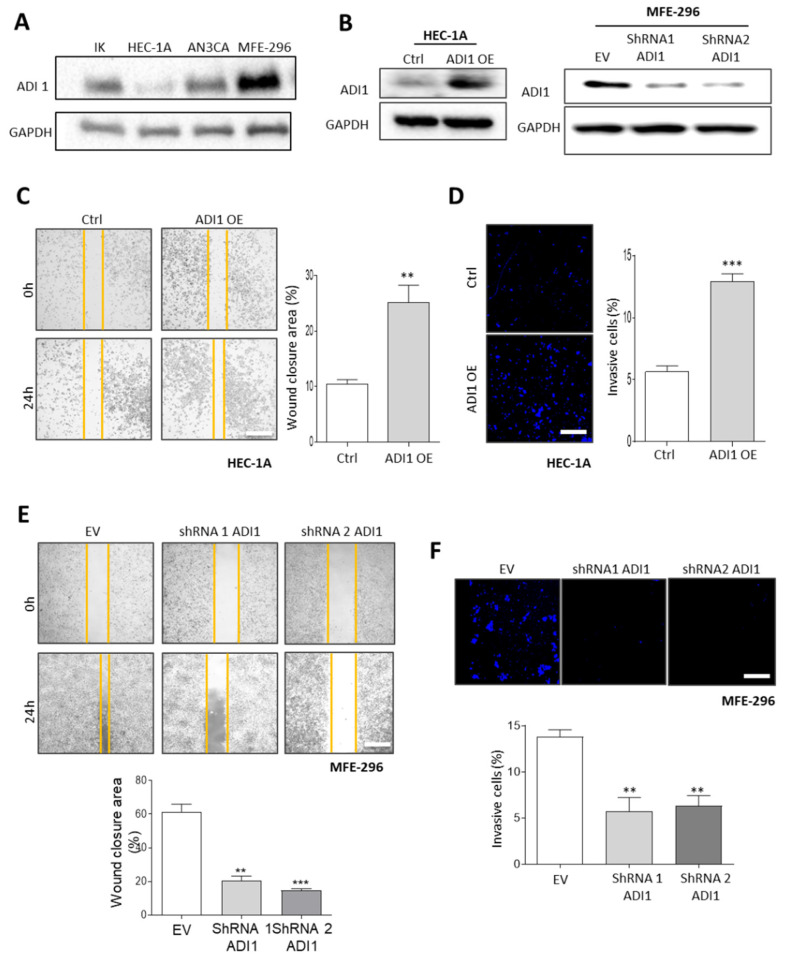
Ectopic modification of ADI1 levels modulates migration and invasion capabilities of EEC cell lines. (**A**) Representative images of western blot analysis of ADI1 levels of Ishikawa (IK), HEC-1A, AN3CA and MFE-296 cells. GAPDH was used as a loading control. (**B**) Representative images of western blot analysis of ADI1 levels (left) on HEC-1A cell line transfected with ADI1 over-expression vector and (right) MFE-296 cells infected with lentiviral shRNA-ADI1 (shRNA1 ADI1 and shRNA2 ADI1) plasmids. GAPDH was used as a loading control. (**C**) Representative images at time markers 0 and 24 h after scratch of wound-healing assay performed in HEC-1A cells transfected with ADI1 over-expression vector (left) and quantification of wound closure area between the indicated times (Right). (**D**) Representative images of nuclear Hoechst staining of transwell invasion assay after the cotton swab in HEC-1A cells transfected with ADI1 over-expression vector (left) and quantification of Matrigel^®^ invasive cells (right). Scale bars: 50 μm. (**E**) Representative images at time markers 0 and 24 h after scratch of wound-healing assay performed in MFE-296 cells infected with lentiviruses carrying shRNA against ADI1 (shRNA1 ADI1 and shRNA2 ADI1) (top panel) and quantification of wound closure area between the indicated times (bottom plot). (**F**) Representative images of nuclear Hoechst staining of transwell invasion assay after the cotton swab in MFE-296 cells infected with lentiviruses carrying shRNA against ADI1 (shRNA1 ADI1 and shRNA2 ADI1) (upper panel) and quantification of Matrigel^®^ invasive cells (bottom plot). Scale bars: 50 μm. Graph values are the mean, and error bars are represented as mean ± S.E.M. Statistical analysis was performed using one-way ANOVA analysis followed by the Tukey’s multiple comparison test or by two-tailed paired Student *t*-test analysis. ** *p* < 0.01; *** *p* < 0.001. Results shown are representative of at least three independent experiments with a minimum of three technical replicates per experiment. The whole western blots were shown in Appendix A.

**Figure 4 cancers-14-02842-f004:**
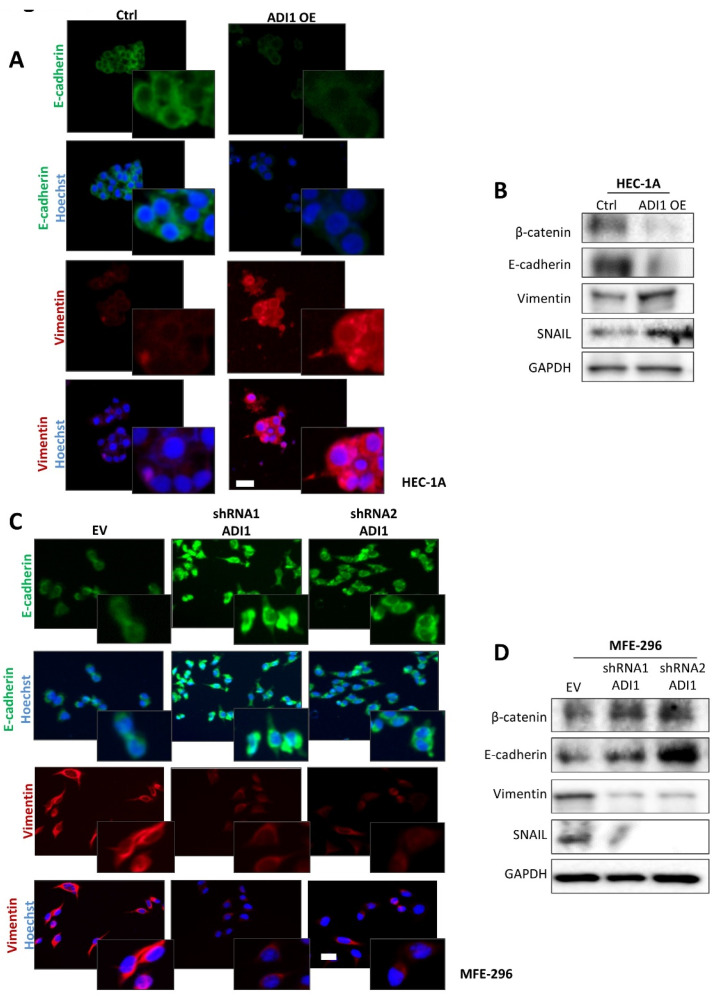
Ectopic modification of ADI1 levels modulates epithelial features. (**A**) Representative images of immunofluorescence against E-cadherin and vimentin in HEC-1A cells transfected with ADI1 over-expression vector. Magnification images of framed regions of the samples are shown. Scale bars: 20 μm. (**B**) Representative images of western blot analysis of E-cadherin, β-catenin, vimentin and SNAIL in HEC-1A cell line transfected with ADI1 over-expression vector. GAPDH was used as a loading control. (**C**) Representative images of immunofluorescence against E-cadherin and vimentin in MFE-296 cells infected with lentiviral shRNA1 ADI1 and shRNA2 ADI1 plasmids. Magnification images of framed regions of the samples are shown. Scale bars: 20 μm. (**D**) Representative images of western blot analysis of E-cadherin, β-catenin, vimentin and SNAIL in MFE-296 cells infected with lentiviral shRNA1 ADI1 and shRNA2 ADI1 plasmids. GAPDH was used as a loading control. The whole western blots were shown in Appendix A.

**Table 1 cancers-14-02842-t001:** Clinical features of human samples used for metabolomics analysis.

Case	Age At Diagnosis	Histological Type	Grade	Stage	Molecular Classification	Recurrence	Follow Up	Duration Follow Up (Years)
1	79	EEC	II	pT3a	MSI	YES	DOD	4
2	78	EEC	I	pT1a	LCN	NO	NED	4
3	56	EEC	II	pT1a	MSI	NO	NED	4
4	69	EEC	II	pT1a	MSI	NO	NED	4
5	85	EEC	II	pT1a	LCN	NO	NED	4
6	79	EEC	II	pT1b	LCN	NO	NED	4
7	76	EEC	II	pT1a	LCN	NO	NED	4
8	60	EEC	I	pT1b	LCN	NO	NED	4
9	65	EEC	II	pT3c	LCN	YES	DOD	4
10	84	EEC	III	pT1b	LCN	NO	NED	3
11	71	EEC	III	pT1b	POLE	NO	DAD	3
12	64	EEC	I	pT1a	LCN	NO	NED	3
13	85	EEC	II	pT1b	MSI	NO	DAD	2
14	86	EEC	I	pT1b	MSI	NO	NED	5
15	80	EEC	II	pT1b	MSI	NO	DAD	5
16	70	EEC	I	pT1a	MSI	NO	NED	6
17	45	EEC	II	pT1a	POLE	NO	NED	7
18	39	EEC	II	pT3c	LCN	NO	NED	7
19	86	EEC	I	pT1a	LCN	NO	NED	7
20	64	EEC	II	pT1b	MSI	NO	NED	7
21	81	SC	III	pT3a	HCN	YES	DOD	7
22	91	SC	III	pT2	HCN	NO	DAD	7
23	76	SC	III	pT1a	HCN	YES	DOD	5
24	81	SC	III	pT3b	HCN	YES	DOD	4
25	58	SC	III	pT3c	LCN	YES	DOD	4
26	83	SC	III	pT3a	HCN	YES	DOD	4
27	74	SC	III	pT3a	HCN	YES	DOD	4
28	NA	SC	III	NA	HCN	NA	NA	NA
29	NA	SC	III	NA	HCN	NA	NA	NA
30	NA	SC	III	NA	HCN	NA	NA	NA
31	76	SC	III	pT3b	HCN	NA	NA	NA

Abbreviations: EEC: endometrioid endometrial carcinoma; SC: serous carcinoma; POLE: exonuclease domain of the DNA polymerase epsilon (group 1); MSI: microsatellite instability (group 2); CNL: low copy-number (group 3); CNH: high copy-number or serous-like (group 4); DOD: death of disease; NED: evidence of disease; DAD: dead of another disease; NA: not assessed.

**Table 2 cancers-14-02842-t002:** Clinical features of human samples used for the immunohistochemical study.

Case	Age at Diagnosis	Histological Type	Grade	Stage	Molecular Classification	Recurrence	Follow Up	Duration Follow Up (Years)
1	80	EEC	II	IB	MSI	NO	NED	3
2	59	EEC	I	IA	LCN	NO	NED	4
3	55	EEC	I	IA	MSI	NO	NED	4
4	54	EEC	III	IA	LCN	NO	NED	3
5	62	EEC	I	IA	LCN	NO	NED	3
6	72	EEC	I	IA	LCN	NO	NED	3
7	63	EEC	I	IA	LCN	NO	NED	3
8	54	EEC	I	IA	MSI	NO	NED	3
9	64	EEC	II	IA	LCN	NO	NED	4
10	56	EEC	I	IA	MSI	NO	NED	3
11	60	EEC	II	IA	MSI	NO	NED	3
12	60	EEC	III	IB	MSI	NO	NED	3
13	69	EEC	I	IA	MSI	NO	NED	3
14	66	EEC	II	IIIB	LCN	NO	NED	3
15	59	EEC	I	IA	MSI	SI	AWD	3
16	80	EEC	I	IB	LCN	NO	NED	3
17	78	EEC	I	IIIB	MSI	NO	NED	3
18	68	EEC	III	IB	MSI	NO	NED	3
19	50	EEC	I	IA	LCN	NO	NED	3
20	80	EEC	I	IA	MSI	NO	NED	3
21	84	EEC	I	IB	MSI	NO	NED	3
22	61	EEC	I	IA	LCN	NO	NED	3
23	86	EEC	II	IA	LCN	NO	NED	3
24	86	EEC	I	II	MSI	NO	NED	3
25	57	EEC	III	IB	MSI	NO	NED	3
26	69	EEC	II	IA	LCN	NO	NED	3
27	69	EEC	I	IA	POLE	NO	NED	3
28	66	EEC	I	IIIC	MSI	NO	NED	2
29	63	EEC	I	IA	MSI	NO	NED	2
30	76	EEC	II	IB	LCN	NO	NED	2
31	62	EEC	I	IA	MSI	NO	NED	2
32	64	EEC	III	II	P53	NO	NED	5
33	53	EEC	III	II	NA	NO	NED	7
34	70	EEC	III	IB	POLE	NO	DAD	7
35	83	EEC	III	IB	LCN	NO	NED	7
36	59	EEC	III	II	MSI	NO	DAD	4
37	79	EEC	II	IA	NA	NO	NED	5
38	88	EEC	II	IA	MSI	NO	DOD	5
39	80	EEC	I	IB	NA	NO	DAD	5
40	69	EEC	II	II	NA	NO	NED	7
41	68	EEC	II	IB	NA	NO	NED	7
42	64	EEC	III	IB	MSI	NO	NED	7
43	77	EEC	I	IB	NA	NO	DAD	6
44	67	EEC	II	IB	NA	SI	AWD	7
45	67	EEC	I	IB	MSI	NO	NED	7
46	49	EEC	III	IV	MSI	NO	DAD	7
47	82	EEC	II	II	NA	NO	DAD	7
48	82	EEC	II	IB	NA	NO	NED	7
49	77	EEC	II	IB	NA	NO	DAD	7
50	75	EEC	II	IB	NA	NO	NED	7
51	75	EEC	II	IB	LCN	NO	NED	7
52	62	EEC	II	II	NA	NO	DAD	5
53	85	EEC	I	IB	NA	NO	NED	8
54	54	EEC	I	III	NA	NO	NED	8
55	69	EEC	I	IB	NA	NO	NED	8
56	80	EEC	II	IB	NA	NO	DAD	5
57	64	EEC	I	IA	NA	NO	NED	8
58	47	EEC	I	IA	NA	SI	DOD	7
59	67	EEC	I	IA	NA	NO	NED	7
60	46	EEC	I	IA	LCN	NO	NED	7
61	82	EEC	I	IA	NA	NO	DAD	6
62	55	EEC	II	IA	NA	NO	NED	7
63	62	EEC	II	IA	NA	NO	NED	7
64	63	EEC	II	IA	NA	NO	NED	7
65	68	EEC	II	IA	MSI	NO	DAD	6
66	70	EEC	I	IA	LCN	NO	DAD	6
67	70	EEC	I	IA	NA	NO	NED	7
68	68	SC	III	IA	P53	NO	NED	3
69	75	SC	III	IB	P53	NO	NED	3
70	84	SC	III	IB	LCN	NO	DAD	3
71	59	SC	III	IA	P53	NO	NED	3
72	87	SC	III	II	P53	NO	NED	3
73	76	SC	III	IV	P53	SI	AWD	3
74	76	SC	III	IIIA	P53	NA	NA	NA
75	70	SC	III	IV	P53	SI	DOD	3
76	73	SC	III	IIIB	P53	SI	DOD	3
77	67	SC	III	IA	P53	NO	NED	2
78	58	SC	III	IV	P53	SI	DOD	2
79	68	SC	III	IIIC	P53	NO	NED	2
80	77	SC	III	IA	P53	NO	NED	2
81	72	SC	III	IA	P53	NO	NED	5
82	55	SC	III	IV	P53	SI	DOD	5
83	77	SC	III	IV	P53	SI	DOD	5
84	75	SC	III	IIIA	P53	SI	DOD	4
85	84	SC	III	IIIA	P53	SI	DOD	4
86	75	SC	III	IIIB	P53	SI	DOD	4
87	91	SC	III	II	P53	NO	DAD	4
88	70	SC	III	IV	P53	SI	DOD	4
89	78	SC	III	IIIC	P53	SI	DOD	4
90	78	SC	III	IB	LCN	SI	DOD	4
91	72	SC	III	IIIV	P53	SI	DOD	4
92	71	SC	III	IA	P53	SI	DOD	4
93	84	SC	III	IV	P53	SI	DOD	4
94	87	SC	III	II	P53	SI	DOD	5
95	77	SC	III	IIIB	P53	SI	DOD	5
96	86	SC	III	IB	P53	SI	DOD	5

Abbreviations: EEC: endometrioid endometrial carcinoma; SC: serous carcinoma; POLE: exonuclease domain of the DNA polymerase epsilon (group 1); MSI: microsatellite instability (group 2); CNL: low copy-number (group 3); CNH: high copy-number or serous-like (group 4); DOD: death of disease; NED: evidence of disease; DAD: dead of another disease; AWD: alive with disease; NA: not assessed.

**Table 3 cancers-14-02842-t003:** Metabolites statistically different between SC and EEC samples (*p* < 0.05).

Class	Subclass	Compound	m/z	rt	*p* Value	FDR	Regulation (sc vs. eec)
Benzene and substituted derivatives	Benzoic acids and derivatives	Phthalic acid ^a^	149.0255	9.5	0.0096685	0.09476	down
		Diheptyl phthalate ^a^	363.2476	11.5	0.0041352	0.059558	down
		Phthalic acid mono-2-ethylhexyl ester ^b^	279.1545	9.5	0.003451	0.058653	up
Carboxylic acids and derivatives	Amino acids, peptides and analogues	Glutathione ^b^	308.0837	0.4	0.040854	0.19345	up
		L-Isoleucine ^a^	249.1468	0.4	0.045153	0.20275	up
		Valine ^a^	118.0784	0.4	0.033932	0.17334	up
		N-Jasmonoylisoleucine ^b^	324.2122	9.5	0.045153	0.20275	up
		Leucyl-phenylalanine ^b^	279.1661	9.5	0.0065027	0.078944	down
Fatty acyls	Fatty acids and conjugates	2-oxo-4-methylthiobutanoic acid ^a^	149.0165	11.9	0.02082	0.13665	up
		Tetracosatetraenoic acid ^a^	343.2939	7.4	0.0009416	0.030514	down
		Eicosatrienoic acid ^a^	307.2528	11.9	0.0087278	0.090507	up
		FAHFA(16:0/13-O-16:0) ^a^	549.4234	12.9	0.010143	0.09476	down
	Eicosanoids	8-iso-15-keto-PGE2 ^a^	373.2053	7.4	0.010132	0.09476	down
	Lineolic acids and derivatives	alpha-Linolenic acid ^a^	279.2193	11.4	0.026493	0.15008	down
	Fatty acid esters	Linoleyl carnitine ^a^	424.3364	9.8	0.020487	0.13665	up
Glycerolipids	Monoradylglycerols	MG(18:1) ^b^	357.293	11.3	0.013272	0.11253	down
		MG(18:3) ^b^	375.2435	10.4	0.0056723	0.070026	down
		MG(20:5) ^b^	399.2496	11.7	0.006547	0.078944	down
		MG(22:0) ^b^	453.3387	5.9	0.011262	0.1022	down
		MG(22:6) ^b^	385.2849	10.4	0.0021001	0.042052	down
		MG(i-17:0) ^b^	367.2718	11.4	0.026493	0.15008	down
	Diradylglycerols	DG(34:4) ^b^	589.4824	13.2	0.00064951	0.021727	down
		DG(36:6) ^b^	613.4706	13.1	0.013272	0.11253	down
		DG(44:11) ^b^	737.4961	10.9	0.033685	0.17334	down
		DG(37:6) ^b^	627.4929	13.2	0.025968	0.15008	down
		DG(36:5) ^b^	615.4996	13.2	0.017212	0.12309	down
		DG(40:4) ^b^	673.5836	15.4	0.024363	0.1494	down
		DG(40:8) ^b^	665.5056	13.2	0.015826	0.12123	down
		DG(i-28:0) ^b^	535.4314	12.9	0.0056723	0.070026	down
		DG(i-33:0) ^b^	583.5331	14.1	0.022169	0.13943	down
Glycerophospholipids	Glycerophosphates	LysoPA(18:1) ^b^	419.2675	10.5	0.0000217	0.0037419	down
		LysoPA(P-16:0) ^b^	377.2554	10.9	0.044511	0.20275	down
		LysoPA(20:3) ^b^	499.2271	6.9	0.013508	0.11253	down
		LysoPA(22:1) ^b^	475.32	8.2	0.0041352	0.059558	down
		LysoPA(20:2) ^b^	463.2957	10.6	0.00027798	0.012533	down
		PA(P-34:2) ^b^	657.503	13	0.02976	0.16074	down
		PA(36:6) ^b^	693.4684	10.8	0.0056723	0.070026	down
		PA(28:1) ^b^	573.4021	12.6	0.025604	0.15008	down
		PA(29:1) ^b^	605.4157	10.7	0.0021604	0.042052	down
		PA(30:2) ^b^	617.4202	10.8	0.00015155	0.0098224	down
		PA(40:7) ^b^	747.4967	12.5	0.021704	0.13943	up
		PA(36:7) ^b^	691.4537	10.5	0.0037003	0.059558	down
		PA(38:5) ^b^	705.4727	10.9	0.0076209	0.084073	down
		PA(44:9) ^b^	781.5217	10.9	0.019748	0.13665	down
		PA(48:0) ^b^	855.7351	14.9	0.0000866	0.0081641	down
		PA(46:7) ^b^	831.5771	12.2	0.0076209	0.084073	down
		PA(48:1) ^b^	853.7199	14.6	0.0014506	0.035815	down
		PA(48:2) ^b^	891.6926	14.3	0.0098617	0.09476	down
		PA(i-27:0) ^b^	561.3887	10.7	0.00097426	0.030615	down
	Glycerophosphoglycerols	LysoPG(18:1) ^b^	493.3069	10.2	0.0000217	0.0037419	down
		PG(40:5) ^b^	825.5453	10.9	0.0029877	0.052513	down
		PG(40:8) ^b^	801.4891	10.6	0.0000217	0.0037419	down
		PGP(40:4) ^b^	889.5384	10.7	0.0028316	0.052435	down
		PGP(i-30:0) ^b^	757.4626	10.6	0.0000433	0.0064146	down
		PGP(i-37:0) ^b^	873.5459	11.2	0.033685	0.17334	down
		PG(38:7) ^b^	793.5243	11	0.022018	0.13943	down
		PG(i-30:2) ^b^	713.4383	10.5	0.0000217	0.0037419	down
		PG(i-28:0) ^b^	649.4432	10.8	0.0005173	0.021458	down
		PG(42:9) ^b^	845.5149	10.7	0.0012528	0.03453	down
	Glycerophosphoinositols	LysoPI(18:2) ^b^	579.2943	9.5	0.003452	0.058653	down
	Glycerophosphocholines	LysoPC(20:1) ^b^	532.3782	10.3	0.00064951	0.021727	down
		LysoPC(22:2) ^a^	576.4025	10.4	0.0021001	0.042052	down
		LysoPC(P-18:1) ^a^	488.3517	10.2	0.022018	0.13943	down
		LysoPC(28:1) ^b^	644.487	10.8	0.045361	0.20275	down
		PC(34:5) ^b^	752.5083	10.6	0.0041352	0.059558	down
		PC(38:6) ^a^	806.5721	13.5	0.035678	0.17618	down
	Glycerophosphoethanolamines	LysoPE(24:1) ^b^	546.392	10.7	0.010132	0.09476	down
		LysoPE(O-16:3) ^a^	434.252	10.7	0.017016	0.12309	down
		PE(O-35:6) ^a^	708.481	10.5	0.0021001	0.042052	down
		PE(38:1) ^a^	774.6007	13.7	0.0035067	0.058653	down
		PE(36:1) ^a^	746.5744	13.5	0.0044838	0.062834	down
		PE(42:9) ^b^	796.534	10.6	0.01991	0.13665	down
	Glycerophosphoserines	PS(38:3) ^b^	814.5753	9.5	0.0028304	0.052435	down
		PS(40:1) ^b^	884.5858	10.7	0.049943	0.22133	down
	Glycerophosphoinositol phosphates	PIP2(36:1) ^b^	1007.5233	10.6	0.02082	0.13665	up
	CDP-glycerols	CDP-DG(38:4) ^b^	1068.5164	13.3	0.045361	0.20275	up
Organooxygen compounds	Carbonyl compounds	Kynurenine ^a^	209.0856	0.8	0.027929	0.15324	up
Prenol lipids	Diterpenoids	Pristanic acid ^b^	299.2912	12.4	0.0041352	0.059558	down
		3-hydroxypristanic acid ^a^	297.2762	12.1	0.00015155	0.0098224	down
Purine nucleotides	Purine deoxyribonucleotides/purine ribonucleotides	2’-Deoxyguanosine 5’-monophosphate/adenosine monophosphate ^a^	348.0631	0.6	0.0021898	0.042052	up
Sphingolipids	Phosphosphingolipids	SM(d41:1) ^a^	801.6941	14.5	0.0041352	0.059558	down
		SM(d42:1) ^a^	797.6668	14.1	0.034987	0.17359	down
		SM(d43:2) ^a^	827.7039	14.6	0.0000217	0.0037419	down
		SM(d40:1) ^a^	787.6773	14.4	0.0056723	0.070026	down
		SM(d42:0) ^a^	839.7151	14.5	0.0002598	0.012246	down
		SM(d44:2) ^a^	841.7301	14.7	0.0002598	0.012246	down
		SM(d41:2) ^a^	799.6723	14.3	0.0002598	0.012246	down
		SM(d40:2) ^a^	785.6655	14.2	0.0076209	0.084073	down
		SM(d42:2) ^a^	813.6951	14.4	0.0056723	0.070026	down
		SM(d43:1) ^a^	829.7289	14.8	0.017212	0.12309	down
Steroids and steroid derivatives	Bile acids, alcohols and derivatives	Ketodeoxycholic acid ^a^	391.2796	11.9	0.026276	0.15008	up
		Chenodeoxycholic acid 3-sulfate ^b^	473.2704	11.7	0.0076209	0.084073	down

All compounds are putatively annotated compounds based upon physicochemical properties and/or spectral similarity with public/commercial spectral libraries [29]. (^a^) Identity (ID) based on exact mass, retention time (RT) and MS/MS spectrum (high fiability); (^b^) ID based on exact mass and RT.

## Data Availability

The datasets analysed during the current study are available from the corresponding author upon reasonable request.

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
