# Peer review of "Metabolomic Analysis Points to Bioactive Lipid Species and Acireductone Dioxygenase 1 (ADI1) as Potential Therapeutic Targets in Poor Prognosis Endometrial Cancer"

_cancers, 2022, doi:10.3390/cancers14122842_

Round 1

Reviewer 1 Report

While the manuscript has been improved in some areas in this version, there are still some points that need be raised.

From Specific point 1:

The nomenclature is again not corrected in all the areas of the manuscript. In addition, some sentence structures have to be corrected (i.e. line 329-330). 

From Specific point 6:

In the response to the specific points the authors introduce a reference (Muinelo-Romay  et al., 2011; https://doi.org/10.1158/1535-7163.MCT-10-1019 ) which cannot be found in the newly submitted manuscript.  Instead another reference has been introduced.

Specific Point (new)

Figure 4 immunoflurescence images need to be enhanced (made bigger).

Author Response

Reviewer-1

While the manuscript has been improved in some areas in this version, there are still some points that need be raised.

We appreciate the time and effort in reviewing our manuscript and your constructive comments. As reflected in this short report, we have addressed all issues suggested.

From Specific point 1:

The nomenclature is again not corrected in all the areas of the manuscript. In addition, some sentence structures have to be corrected (i.e. line 329-330). 

We have revised the nomenclature used throughout the manuscript. In addition, we have rewritten the entire paragraph, from 330 to 346 lines, in order to clarify the rationing carried out during the experiments.

 From Specific point 6:

In the response to the specific points the authors introduce a reference (Muinelo-Romay  et al., 2011; https://doi.org/10.1158/1535-7163.MCT-10-1019 ) which cannot be found in the newly submitted manuscript.  Instead another reference has been introduced.

We apologise for the error. We have introduced properly the reference ([31]).

Specific Point (new)

Figure 4 immunoflurescence images need to be enhanced (made bigger).

As suggested, we have enhanced immunofluorescence images

Reviewer 2 Report

The authors of this manuscript have done a rigorous review and made the suggested changes, so in my view, the manuscript has been improved and can be published in this form. However, it would be convenient that both Table I and especially Table II should have a smaller presentation, occupying less space within the publication. Likewise, the visualization of Figure 1B (Hierarchical clustering of individual samples according to metabolite levels) continues to be so unclear that it could be passed to "Supplementary Material", since seeing it does not contribute much.

Author Response

The authors of this manuscript have done a rigorous review and made the suggested changes, so in my view, the manuscript has been improved and can be published in this form. However, it would be convenient that both Table I and especially Table II should have a smaller presentation, occupying less space within the publication. Likewise, the visualization of Figure 1B (Hierarchical clustering of individual samples according to metabolite levels) continues to be so unclear that it could be passed to "Supplementary Material", since seeing it does not contribute much.

We really appreciate the time and comments of the referee.

As suggested, we have been modified all issues suggested regarding tables size and Figure 1B (which has been moved to Supplementary figure-1)

Reviewer 3 Report

Page 6/23 please provide an adequate reference that FFPE blocks older than 2 years should not be used for IHC, I don`t quite agree with this time range.

In Table 1+2 it says follow up but I don`t find any details about the duration of follow up and, how NED etc was measured in the text or table legends.

Fig 1 please increase quality of the figures A + C

Fig 2 please provide magnification etc details in IHC pictures/legend

Fig 2 C/D plots have a different size

Fig 2 provide same quality in E/F as in B

English style and grammar changes need to be done – particularly in the parts that are highlighted in yellow.

Author Response

We very much appreciate the reviewer's critical yet constructive comments, allowing us to reassess and improve our manuscript. Therefore, we have revised all the issues suggested.

Page 6/23 please provide an adequate reference that FFPE blocks older than 2 years should not be used for IHC, I don`t quite agree with this time range.

In the literature there are many references regarding the use of old FFPE blocs. The time range is variable depending on the series and the antibodies used. In our experience, we find the best immunoexpression in recent FFPE (not older than 2 years) especially with antibodies not ready to use. As we can find different years range in literature, we have modified the text and provide some references (Page 6/23, lines 258-259).

In Table 1+2 it says follow up but I don`t find any details about the duration of follow up and, how NED etc was measured in the text or table legends.

We have included a column in table were we show the follow up duration of each patient and we have defined the follow up criteria in Materials and Methods section.

Fig 1 please increase quality of the figures A + C

We agree with reference’s comment. For that reason, we have provided better quality figures 1A and 1B (previously Fig 1C)

Fig 2 please provide magnification etc details in IHC pictures/legend

We have provided the magnification information in the figure legend. 

Fig 2 C/D plots have a different size

As suggested by the reviewer, we have re-sized figure 2C and D plots.

Fig 2 provide same quality in E/F as in B

We agree with reference’s comment. For that reason, we have provided same quality figures.

English style and grammar changes need to be done – particularly in the parts that are highlighted in yellow.

We have revised and modified the English style and grammar. Special attention has been paid to the areas highlighted in yellow.

This manuscript is a resubmission of an earlier submission. The following is a list of the peer review reports and author responses from that submission.

Round 1

Reviewer 1 Report

The manuscript by Gatius et al uses metabolomics profiling analysis to define the specific metabolomic signature of uterine serous carcinoma (SC). Out of the 232 statistically different metabolites among SC and Endometrioid EC (EEC) patient samples, most were lipid species with  lower levels in SC, and amino acids and purine nucleotides with higher levels in SC. The authors examined the role of ADI1 enzyme, which synthesizes 2-Oxo-4-methylthiobutanoic acid. Levels of ADI1 were higher in SC compared to EEC, and ADI1 mRNA levels were higher in p53 abnormal ECs compared to p53 wild type tumours, showing a statistically significant negative correlation with overall survival and progression-free survival among EEC patients. ADI1 was also shown to induce migration and invasion in EC cell lines.

Specific point 1:

  1. The quality of writing and data presentation needs improving. The misspelling of several phrase, percentages given (eg. line 40 vs line 95) and parts of sentences, and more notably the metabolite fatty acid 2-Oxo-4-methylthiobutanoic acid throughout the text (line 46, 59, 336, 497) discourages from reading.

In addition, data presented in current form is non-legible, the fonts are too small, and therefore the intended data presented is obscured.

Specific point 2:

  1. Line 57 reads 892%, from where does this percentage arise?

Specific point 3:

  1. Entire paragraph regarding patient selection pertaining to Figure 1 should be removed from Materials and methods and placed into Results (line 133-143)

Specific point 4:

  1. Figure 2D is scarcely explained in the Results, moreover it is not clear from where the statement in line 292-293 reading ‘Interestingly, 13/15 metabolites were down-regulated in SC samples. ’ arises.

Specific point 5:

  1. The entire 3.1. segment lacks more critical overview of results, although they are presented as part of the discussion, they might serve as part of results more to explain the transition to next line of investigation.

Specific point 6:

6.The ms would be stronger had the authors made an attempt to identify a mechanism of action of ADI1 which contributes to the observed malignancy potential.

Reviewer 2 Report

In their manuscript, the authors analysed diffrences in metabolomic profiles between subgroups od EC, trying to identify differences between serous (SC) and endometroid (EEC). They performed a comprehensive analysis copuled with bioinformatic tools showing that the two entities indeed have a different metabolomic profile.

However, the EEC group had different histological grades and stages compared to the high grade of SC. No mention of eventual differnce s depending on the grade or stage are reported. Comparing high grade EEC and SC can the authors find the same results?

It is not clear how the authors embarked in ADI1 from all the data retried from the metabolomics analysis. Dod they try several without finding differences? did they pick it by chance?  I was wondering wihch results could be found selecting other metabolites differently expressed.

The data on correlation between OS, PFS and expression are of limited value if not reproduced in prospectic studies. We have several examples of potenital biomarker found in public databases then not confirmed.......

In addition it seems to me that n SC whatever the role of these analysis there is a difference between OS and PFS in SC. IN one case ther is better PFS for low expressing patients, but the contrary is found for OS. Is there any explanation? does this reflect the poor statistical power and clinical significance of these analysis?

The correlation with p53 is also difficult to understand from the data. I wonder, again if other random correlation can be found staistically signficiant, being the data not so titillating.

Finally the data reported in cell lines are encouraging but very preliminar. What happens in cells overexpressing ADI1 following transfection or in those in which ADi1 is reduced by siRNA in terms of growth? What happens in vivo? These are in my opinion information that can indicate this as a potential target.

In conclusion, the authors started aiming at finding differneces betweeb SC and EEC (trying to justufy the poor prognosis of SC compared to EEC) and ended with a preliminary characterizaion of one protein in EEC that is far from being satisfactory. I think more data are needed to propse it as a tagret as well as a biomarker of aggressivness

Reviewer 3 Report

-In the Introduction the expression "aggressive histology" I think is not appropriate and can be replaced by "histological type" or "histological severity degree".
-In Results, the description of the "lipid species" appears boring, taking into account that everything is in Table 1, perhaps it could be limited, only highlighting the interesting last 5 lines.
- Figures 1A and 1B neither provide nor help improve a global view of the parameters of the human samples studied here. The series of small colored bars or squares do not particularly help to better visualize a description of: grade, stage, Molecular Classification, recurrence, or follow up. Nor do Figures 2E and 2F provide a very clear overview of what is described in the text.
-The conclusion that changes in metabolism observed in tumor cells may provide useful information "to monitor treatment and propose therapeutic targets" is not based on findings from the current study

Reviewer 4 Report

The presented study investigates metabolic differences in two different subtypes of EC. The authors not only characterize the metabolomic profile of serous and endometrioid histology but also focus in detail on one of the identified metabolites synthesized by ADI1. Even though the patient number of the metabolomic profiling is very limited, detailed further molecular biological information is given on the potential role of ADI1 in EC. The authors well identified a gap in knowledge on metabolomics signatures and characteristics of cancer subtypes.

Major

The patient number of the cohort used for metabolomic profiling is very low,

compared to a larger cohort of IHC samples; please clarify the origin of the two cohorts and why you selected only this limited number of samples for the metabolomics analyses. Can you provide data of paired samples of metabolomic profiling and IHC analyses?

The figures should be provided in better quality.

Moderate English changes are required.

Minor

Abstract

Please integrate patient numbers in the abstract

Check for spelling and typing errors.

Material and Methods

Patient selection

The patient number is not clear, is it  n=24 EC or EEC for metabolomics?

Various cohorts are used for different analyses, please provide a more detailed information on how the respective patients were selected and how many different patient collectives you used.

It says in 2.1 24 samples (Fig1A) and in Fig1A only 19 samples are depicted – this needs to be clarified

Fig 1B it`s impossible to read how many patients are depicted due to quality issues

IHC

what kind of “non-tumor” samples were used?

Manuscript

Throughout the manuscript the authors say …lower levels “of” SC –this should be corrected to “in”

Page 8/line 324

After Fig2 it says Table 1 followed by a sentence that is not clear where it belongs to (main text?, Table?)

Page 9 line 335 – instead of both “carcinomas” the authors should refer to subtypes (also throughout the rest of the manuscript)

Page 9 line 353 rewrite the sentence

Figure 3 indicate included sample sizes, respectively

Please provide a better quality of fig 3 C and D

In fig 4 D (left) it says EV but (right) it says Ctrl quantification – please clarify

Please improve image quality!

Page 12 line 418 – this phrase should be part of the discussion not result section

Discussion

First paragraph is very similar to the introduction, please abbreviate